# Food Insecurity during the Pandemic in South Korea: The Effects of University Students’ Perceived Food Insecurity on Psychological Well-Being, Self-Efficacy, and Life Satisfaction

**DOI:** 10.3390/foods12183429

**Published:** 2023-09-14

**Authors:** Yoojin Lee, Hyehyun Yoon, Taehee Kim, Hyosun Jung

**Affiliations:** 1Smart Education Platform, Kyung Hee University, 26 Kyungheedae-ro, Dongdaemun-gu, Seoul 02447, Republic of Korea; yoojinlee90@khu.ac.kr (Y.L.); thkim33@khu.ac.kr (T.K.); 2Department of Culinary Arts and Foodservice Management, College of Hotel & Tourism Management, Kyung Hee University, 26 Kyungheedae-ro, Dongdaemun-gu, Seoul 02447, Republic of Korea; hhyun@khu.ac.kr; 3Center for Converging Humanities, Kyung Hee University, 26 Kyungheedae-ro, Dongdaemun-gu, Seoul 02447, Republic of Korea

**Keywords:** food insecurity, psychological well-being, self-efficacy, life satisfaction, university student

## Abstract

This study examined the impact of university students’ perceptions of food insecurity on psychological well-being, self-efficacy, and life satisfaction and observed that the students’ gender plays a moderating role in this causal relationship, based on a total of 491 university students who participated in this empirical study. This study used SPSS (Version 22.0) and AMOS (Version 20.0) for the analyses. This study examines the structural relationship of this causal model. Our findings suggest that students’ perceived food insecurity negatively affects the status of their psychological well-being and self-efficacy. However, contrary to expectations, perceived food insecurity has no negative effects on students’ life satisfaction. In addition, the level of students’ psychological well-being positively influences their life satisfaction, while self-efficacy does not. The moderating effects of gender differences in this research were also disclosed. Limitations and future research directions are also discussed.

## 1. Introduction

The global food environment is currently facing serious crises because of various external factors arising from environmental and social problems [1,2]. In particular, the COVID-19 pandemic, an unpredictable outbreak of infectious diseases, has caused diverse barriers that restrict people’s basic quality of life [3,4] and has brought significant negative impacts on global food security [5,6]. On account of quarantine and border closures caused by COVID-19, food exchange and trade between countries have been restricted, resulting in higher food prices and lower food accessibility [7,8,9]. Today’s rapid changes in the food environment are changing the way people consume and choose food, resulting in side effects that seriously threaten people’s eating habits [10,11,12,13,14]. It is expected that the broken balance between global food supply and demand will further exacerbate people’s food safety and dietary diversity in the years ahead [15,16,17].

As the food environment has become increasingly unstable with the progression of severe crises, the food insecurity problem has currently intensified at an alarming level [8,18,19]. In general, food security, defined as “people’s physical, social and economic access to sufficient, safe and nutritious food that meets their dietary needs and food preferences for an active and healthy life”, is based on four important pillars, namely the availability, accessibility, utilization, and stability of food [20]. These multidimensional aspects of food security are recognized as basic human rights to be guaranteed to all citizens [21]. In contrast to the notion of food security, food insecurity is most widely defined as “limited or uncertain availability of nutritionally adequate and safe foods or limited or uncertain ability to acquire acceptable foods in socially acceptable ways” [22]. It implies a situation in which an individual has difficulty obtaining food and suffers from nutrition deficiency with mental distress [23,24,25]. For this reason, food insecurity problems are perceived as severe threats to people’s daily lives, and they even result in many terrible diseases that not only affect physical aspects but also psychological well-being [26,27]. Furthermore, many recent research studies have confirmed that food insecurity issues are closely related to the level of one’s self-efficacy and life satisfaction [28]. According to previous studies, people who were brought up in food-secure and affluent households generally act in an autonomous manner with firm confidence, and they feel less restricted in attempting new challenges [29,30]. They can cultivate self-esteem through continuous processes of trial and error, and they intend to seize opportunities for developing self-efficacy [31,32]. Thus, food security is regarded as one of the primary assets that can enhance one’s psychological well-being and self-efficacy level, with a major role in improving overall satisfaction [33,34,35].

In the past, food insecurity, which has a profound impact on one’s life, was a socially problematic issue normally only found in disadvantaged households [36,37,38]. Today, however, as many environmental and social incidents that threaten stable food systems constantly occur around the world [39], the general public is also faced with a high vulnerability to the food insecurity problem [3,40]. In particular, university students who become an adult for the first time with unfamiliar autonomous discretion are recognized as the most economically and socially marginalized [41,42,43,44], and they are in danger of the high prevalence of food insecurity these days [14]. Although healthy eating habits need to be formed during that period, today’s university students often purchase relatively cheap and unhealthy ultra-processed foods, since they have no fixed income and have high living costs [45,46,47,48]. In addition, because they are swamped with studying and workloads, they do not have enough time to acquire fresh and healthy food regularly [49,50] and are eventually exposed to the risk of food insecurity, which becomes a serious threat to their health and even academic success [51,52]. Accordingly, unhealthy food consumption behavior is most observed in this generation of undergraduates, especially with respect to students who come from low-income and food-insecure families [53,54,55].

Since the eating habits of university students are the primary determinants that influence their entire lifetime’s eating habits [56,57,58] and the overall quality of life in the long run [59,60,61,62], it is necessary to examine the present status of students’ food insecurity level and also its impacts on psychological aspects that are related to the satisfaction of life [63,64,65]. However, most studies so far have focused on the impact of food insecurity on physical consequences such as nutritional status or physical illness. Moreover, there is little research that goes into great depth on the relationship between food insecurity and psychological or cognitive domains. In addition, despite the rapidly changing food environment after the COVID-19 pandemic, there is no research focusing on the food insecurity and psychological states of university students in Republic of Korea during the pandemic era. Therefore, existing research studies are insufficient for explaining the complex relationship of the research variables above; furthermore, intensive investigation should be carried out to clarify the exact influences on food insecurity. Under this research background, this study conducted a profound analysis to explore the relationships between food insecurity, psychological well-being, self-efficacy, and life satisfaction among university students in Republic of Korea.

## 2. Literature Review

### 2.1. Food Insecurity

Food insecurity, which means “limited availability of nutritionally adequate foods or uncertain ability to acquire safe foods in socially acceptable ways”, raises severe physical and psychological problems in peoples’ lives in both short and long terms [22,26,66]. In physical aspects, since people in food insecure households face chronic food shortages [67,68,69], they are generally in nutritional imbalance states, which cause health problems such as malnutrition or low blood pressure [52,70]. In addition, a lot of food insecure people are overly dependent on unprocessed or energy-dense food because of their limited food budget [71,72]. Accordingly, this unhealthy eating pattern leads them to the destruction of their physical condition by inducing various adult diseases such as overweight, obesity, diabetes, and even cardiovascular diseases [73,74].

Not only does food insecurity adversely affect physical health, but also it leads to fatal consequences for psychological and mental well-being [75,76,77]. Since food expense is one of the crucial and indispensable expenditures of everyday life, it is difficult to considerably save or reduce the budget [78,79]. Therefore, people who suffer from food insecurity are under tremendous stress, anxiety, fear, and even suicide ideation with financial pressure [80,81]. Moreover, when people cannot take enough nutrition and fail to satisfy their appetites, they tend to experience psychiatric disorders with severe depression which often manifest in abnormal or self-destructive behavior [82,83,84]. Therefore, there is no doubt that food insecurity causes great negative impacts not only on an individual’s physical and psychological well-being but also the overall quality of life [51,85,86,87]; this also produces negative knock-on effects on the stability of one’s family, community, and society as a whole [88,89,90]. Accordingly, many researchers in various fields are paying great attention to the significant function of food security and the seriousness of food insecurity problem issues around the world [91,92,93]. Governments are also attempting to seek efficient solutions for mitigating these grave problems [94,95,96,97].

### 2.2. Psychological Well-Being

The nature of one’s psychological well-being, which originated from positive psychology discourses, indicates one’s multidimensional, psychological, and spiritual state, which is derived from favorable affects and the perceived feelings of satisfaction in one’s daily life, and it is greatly influenced by various surrounding environmental factors [98]. For this reason, psychological well-being is very difficult to define exactly and is used interchangeably with the terms of subjective well-being, subjective quality of life, and happiness [99]. Ryff [100], one of the distinguished scholars in psychological well-being research, emphasized the eudaemonic aspects of well-being and embodied the concept of psychological well-being through a specific multidimensional model comprising the notions of autonomy, environmental mastery, personal growth, positive relations, purpose, and self-acceptance.

In the context of food and eating, however, psychological well-being has been interpreted in more expanded views including hedonic aspects [101,102]. The reason is that in people’s lives, food is not only responsible for supplying fundamental nutrients, but also has the potential to elevate moods through the pleasure of gastronomy and commensality [103]. Therefore, many studies exploring the relationship between one’s dietary pattern and psychological well-being have been conducted from diverse areas of research. For example, in clinical fields, Barnhart et al. [104] and Chan and Chiu [105] identified that peoples’ disordered eating behaviors are closely connected with their psychological well-being. Moreover, in the studies of Bucchianeri et al. [106] and Ickes et al. [107], the associations between one’s eating habits and psychological health appear differently depending on specific demographic information such as age, gender, or race. Many researchers today especially pay great attention to the topic of students’ psychological well-being related to their food insecurity status. Men and Tarasuk [108] and Polsky and Gilmour [109] asserted that a lot of students around the world seriously suffer from psychological problems, which have been derived from food insecurity since the COVID-19 pandemic. Men et al. [110] also found that food insecurity hinders students’ academic success and even causes considerable psychological stresses, which become a real threat to their whole lives.

### 2.3. Self-Efficacy

Self-efficacy, stemming from Bandura’s social cognitive theory, is defined as “one’s belief in their capability to regulate negative emotions and to plan and follow sustainable actions which will bring desired outcomes and achievements” [111]. Bandura [111] discovered the critical function of self-efficacy in people’s lives and disclosed that self-efficacy gives them confidence to change existing behaviors and enhances the possibility of creating better performances in the future [112]. Due to this hopeful and future-oriented nature, the notion of self-efficacy has been treated as significant in the healthcare industry to help patients recover [113], and it also draws a great deal of attention from many researchers for improving public health and welfare to this day [114,115].

In the context of food and eating, consumers’ self-efficacy has become more important in making healthy and safe food choices under an industrialized food system [116]. As Anderson et al. [117] pointed out, individuals who have high levels of self-efficacy and self-confidence are not easily tempted by junk foods and maintain healthy diets with daily exercises. Moreover, many researchers discovered that self-efficacy levels in choosing healthy foods can be greatly enhanced through social support such as families assisting with home meal preparation, financial aid from communities, or health promotion interventions from educational institutions [118,119,120,121]. Furthermore, it was discovered that self-efficacy plays a pivotal role in increasing one’s food security levels, especially when people are under poor surrounding conditions of living or in a terrible environmental crisis [122,123]. Therefore, self-efficacy is closely bound up with one’s eating habits, and it is considered to be a critical factor in determining the quality of their overall dietary life.

### 2.4. Life Satisfaction

Life satisfaction, generally defined as “individual’s subjective evaluation of their overall quality of life”, is closely linked with the ideas of happiness and the achievement of a meaningful life [124,125]. In other words, the level of life satisfaction relies heavily on people’s positive affectivity and perceived quality of life and well-being [126]. Therefore, many researchers in positive psychology consider the satisfaction with life index to be a significant construct for predicting people’s mental health [127] and discovered that not only internal personal factors such as personalities but also diverse external influences such as interpersonal relations, working environments, or media seriously affect people’s satisfaction with their lives [128,129,130].

Food and healthy eating, which are indispensable elements in maintaining one’s life, are significant factors that determine people’s quality of life and satisfaction level [131,132]. For this reason, various research have explored the complex relationship between people’s eating behavior and life satisfaction [133,134]. In a recent study by Hempel and Roosen [135], the associations between the purchase intention of healthy food, locus of control, and life satisfaction was analyzed in detail in the context of COVID-19. Furthermore, Selvamani et al. [136] and Holm et al. [137] discovered that food insecurity is a key obstacle in practicing healthy eating habits, consequently inducing serious negative impacts on people’s life satisfaction.

## 3. Research Hypothesis

According to previous studies, food security is positively related to people’s safe and happy life and negatively related to both their physical and psychological health [20]. This indicates that if people have insufficient resources or inadequate capabilities for sustaining stable diets, people feel threatened and even fear for their whole livelihood at a dangerously high level [81,85]. This tendency toward food insecurity has been stronger than in the past, especially for university students [14,51]. Accordingly, the low level of food security is expected to have serious adverse effects on students’ psychological and mental states and even aggravate their overall life satisfaction.

### 3.1. Relationship between Food Insecurity and Psychological Well-Being

Food insecurity is closely linked to one’s poor health, especially with regard to psychological and mental aspects [27,138]. According to the research studies of Jessiman-Perreault and McIntyre [139] and Men et al. [110], students’ food insecurity has significant adverse effects on their mental stability. Moreover, Bruening et al. [140] and Reeder et al. [82] identified that food insecurity exacerbates students’ psychological disorder problems by severely interfering with their daily functioning in school life. In addition, Men and Tarasuk [108] and Polsky and Gilmour [109] found that many students have been suffering from serious psychological distress derived from food insecurity since the COVID-19 pandemic, and this tendency is particularly stronger in students who were under severe food restraints from financial vulnerability and inaccessibility to healthy foods [141,142]. Therefore, this study assumes that university students’ food insecurity will have a negative influence on their psychological well-being.

**H1.** 
*Food insecurity negatively influences psychological well-being.*


### 3.2. Relationship between Food Insecurity and Self-Efficacy

Not only students’ psychological well-being but also their self-efficacy is closely associated with the notion of food insecurity [28,122,143]. According to Anderson et al. [112] and Martin et al. [35], an individual’s food insecurity situation has a great detrimental effect on psychological aspects as well as cognitive skills such as self-efficacy, self-esteem, and self-confidence. Moreover, as Muturi et al. [132] notes, self-efficacy plays a crucial role in enhancing the food security level of adolescents, especially when they are under poor surrounding conditions of living or in a terrible environmental crisis. Based on these prior findings, similar results were expected in this research on university students, and this study proposes the following hypothesis as follows:

**H2.** 
*Food insecurity negatively influences self-efficacy.*


### 3.3. Relationship between Food Insecurity and Life-Satisfaction

Life satisfaction is another important element that is intertwined with one’s food security status. Chung et al. [144] and Grunert et al. [145] verified serious adverse effects of food insecurity on the life satisfaction of individuals. In addition, in the studies of Salahodjaev and Mirziyoyeva [19] and Holm et al. [137], food insecurity derived from unstable food costs and restrictive access to a safe food environment is a major obstacle in practicing healthy eating diets, consequently inducing serious negative impacts on people’s life satisfaction. Therefore, on the basis of previous research results, this study formulates the following hypothesis to examine the influence of food insecurity on students’ life satisfaction:

**H3.** 
*Food insecurity negatively influences life satisfaction.*


### 3.4. Relationship between Psychological Well-Being and Life Satisfaction

As well as people’s food security status, psychological well-being is closely related to perceived quality of life and overall life satisfaction [125,126,127]. According to the prior findings of Hege et al. [60], one’s psychological well-being has a positive impact on their life satisfaction. In addition, Magallares et al. [146] asserted that psychological well-being and life satisfaction are inextricably linked to each other and should be considered together in people’s daily lives. For these reasons, this study assumes that university students’ psychological well-being will positively affect their life satisfaction level.

**H4.** 
*Psychological well-being positively influences life satisfaction.*


### 3.5. Relationship between Self-Efficacy and Life Satisfaction

Self-efficacy, which refers to the belief that one can perform a specific action to achieve one’s goal, not only plays a role in encouraging action in real life but also sways one’s satisfaction level of their overall quality of life [29]. In the studies of Hempel and Roosen [135] and Muturi et al. [123], self-consciousness such as self-efficacy and self-esteem has a major impact on one’s quality of life by determining life satisfaction levels. Furthermore, Berman [147], Oishi et al. [148], and Proctor and Linley [149] identified self-efficacy as an influential predictor in measuring people’s life satisfaction and perceived quality of life. In this context, this study proposes the following hypothesis:

**H5.** 
*Self-efficacy positively influences life satisfaction.*


### 3.6. Moderating Role of Gender

According to the results of the research by Rajikan et al. [61], male students tend to be more conscious of their food insecurity than female students. Also, Barry et al. [56] identified that males, especially those who are inexperienced in solving their food insecurity status, face more serious difficulties in coping with poor daily diets than females who generally have more knowledge about skillfully managing their eating plans. On the contrary, Becerra and Becerra [150], Bruening et al. [24], and Jung et al. [36] verified that female students are more vulnerable to food insecurity, and they sustain a lot of mental traumas from their precarious living and unstable eating habits. Based on the various results of these preceding studies, it is expected that the moderating effect of gender will be discovered in this research (see Figure 1).

**H6.** 
*Gender moderates the effects of food insecurity on psychological well-being, self-efficacy, and life satisfaction.*


## 4. Methodology

### 4.1. Sample and Data Collection

In this study, university students were selected as subjects, and questionnaires were disseminated and collected using Embrain, a specialized data collection company. The English questions in the original questionnaire were translated into Korean and then translated again into English to confirm that there was no contextual difference between the Korean and English versions [151]. Prior to the main survey, a preliminary survey was administered to 50 subjects. Based on the results of this survey, questions found to be difficult or ambiguous were modified for use in the main survey. In total, 500 questionnaires were distributed over a one-month period from 1–15 October 2022. The respondents were assured that the data collected would remain confidential. In total, 491 questionnaires were used in the final analysis. The survey respondents consisted of 332 females (67.6%) and 159 males (32.4%), with a relatively similar number of students according to grade: first year (*n* = 124, 25.3%), second year (*n* = 126, 25.6%), third year (*n* = 117, 23.8%), and fourth year (*n* = 124, 25.3%).

### 4.2. Measures

In this study, university students’ perceptions of a set of variables (e.g., food insecurity, psychological well-being, self-efficacy, and life satisfaction) were measured. Based on the preliminary survey, 18 questions were selected, and all items were measured on a 7-point Likert scale (not at all to very much). A 7-point scale was selected because the respondents were less likely to be biased toward the median compared to a 5-point scale. In addition, a 7-point scale can more readily identify differences among respondents. Moreover, in this study, each item was measured using multiple questions, because multiple questions per item can measure research concepts more precisely than a single question per item. The general characteristics of the subjects comprised two questions each for gender and grade.

In addition to background information, food insecurity, which is defined as the limited availability of nutritionally safe food or the uncertain ability to obtain food in a socially acceptable way, was measured with six questions with reference to the questions used in a study by Johnson et al. [152]. These included the following two questions: “During the last 12 months, you ate less than you thought you should?” and “You were worried you would not have enough food to eat?” Psychological well-being, a psychological concept that is considered to constitute the quality of an individual’s life, was measured with five statements based on a study by Ryff [98]. The statements used for measuring this item included the following: “In general, I feel I am in charge of the situation in which I live, and I find it satisfying to think about what I have accomplished in life”. Self-efficacy, which refers to the self-evaluation of one’s belief in their ability to reach a goal, was measured with five statements based on studies by Bandura [111] and Schwarzer and Jerusalem [153]. The statements for measuring this item included “I can always manage to solve difficult problems if I try hard enough, and I am confident that I could deal efficiently with unexpected events”. Life satisfaction, which can be defined as the evaluation of a person’s perceived satisfaction with their overall life [154,155], was measured using three statements based on a study by Diener et al. [156] and it included statements, such as “The conditions of my life are excellent, and I am satisfied with my life”.

### 4.3. Statistical Analysis

The collected data were analyzed using Statistical Product and Service Solutions (SPSS) and Analysis of Moment Structures (AMOS). The normal distribution of the measurement items was confirmed, and common-method variance (CMV) was tested using multicollinearity and the Harman test. The validity and reliability of the measurement items were verified through confirmatory factor analysis (CFA) and reliability analysis. The average variance extracted (AVE), composite construct reliability (CCR), average shared variance (ASV), and maximum shared variance (MSV) were calculated to determine the validity of the measurement items [157]. In addition, structural equation modeling (SEM) and multi-group analysis (MGA) were used to test the study’s hypotheses [158].

## 5. Results

### 5.1. Measurement Model

Table 1 shows the reliability and validity analysis results for the measurement items used in this study. An exploratory factor analysis was conducted prior to the reliability and validity analysis to examine the presence or absence of common method bias (CMB). The results were factorized into a total of four factors with an eigen value of 1 or above, and the total explanatory power was 82.341%, which is very good. The first factor’s explanatory power was 35.783, accounting for less than half of the total explanatory power, which confirmed that the level of CMB was very low. Based on these results, CFA was performed. As shown by the results, the model’s goodness of fit was satisfactory, with χ^2^ = 497.979, df = 129, χ^2^/df = 3.860, GFI = 0.894, TLI = 0.929, CFI = 0.940, IFI = 0.940, RMSEA = 0.076, and RMR = 0.054. The standardized regression coefficients of all measurement items were statistically significant (*p* < 0.001), the CCR and Cronbach’s alpha values were 0.7 or above, and the AVE value was 0.5 or above, which was excellent. Therefore, the reliability and validity of the measurement items used in this study met the required criteria [159]. A correlation analysis was conducted to determine whether the direction between the measurement items was consistent with the direction of the hypotheses (Table 2). As shown by the results, both the direction of the hypotheses and the correlation coefficients were consistent.

### 5.2. SEM

In this study, a structural equation analysis was employed to verify the five hypotheses proposed to consider the direct impact of the measurement items (Table 3). The final model’s goodness of fit was relatively good, with χ^2^ = 372.917, df = 127, χ^2^/df = 2.936, GFI = 0.924, IFI = 0.960, CFI = 0.960, RMR = 0.48, and RMSEA = 0.063. Although a better fit could have been obtained using a modified index, this study’s final model was confirmed without using the index in this study to ensure its validity. Hypothesis 1 assumed that university students’ perceptions of food insecurity would have a negative impact on their psychological well-being. This hypothesis was accepted, given that a higher awareness of food insecurity led to a corresponding lower level of psychological well-being (β = −0.129, *t*-value = −2.476, *p* < 0.05). Hypothesis 2 assumed that perceived food insecurity would lower self-efficacy. This hypothesis was also accepted because university students’ food insecurity significantly reduced their self-efficacy (β = −0.144, *t*-value = −2.814, *p* < 0.01). This means that students’ greater anxiety about food and their experience of inadequate food reduced their psychological well-being and self-confidence in their abilities. However, the negative impact of food insecurity on life satisfaction, as set out in Hypothesis 3, was rejected because the impact was not significant (β = −0.029, *t*-value = −0.085, *p* > 0.05). Hypothesis 4, which assumed that university students’ psychological well-being would increase their life satisfaction, was also accepted (β = 0.715, *t*-value = 9.566, *p* < 0.001). This means that a psychologically enjoyable and fulfilling emotional state improves life satisfaction. Finally, the positive impact of self-efficacy on life satisfaction, which was set out in Hypothesis 5, was rejected because this impact turned out to be insignificant (β = 0.080, *t*-value = 1.189, *p* > 0.05).

### 5.3. Moderating Effect of Gender

To verify the moderating role of gender (Hypothesis 6) in the relationship between food insecurity and psychological well-being, self-efficacy, and life satisfaction, a multi-group analysis was performed using the difference in the degree of freedom between the free and constraint models (Table 4). Given that the difference in the degree of freedom between these models was 1, a significant moderating effect can be assumed if the difference in the chi-square value is greater than 3.84. Before verifying the moderating effect, we analyzed measurement invariance according to gender. As shown by the results, there was no significant difference in the invariance model. Therefore, it was also confirmed in advance that there was no problem with measurement invariance [160]. As a result of the study’s analysis, a gender-specific moderating effect was found in the negative impact of food insecurity perceived by university students on their psychological well-being and self-efficacy. To be specific, the negative impact of food insecurity on psychological well-being was significantly greater among male than female students (male students: β = −0.307, *t*-value = −3.347, *p* < 0.001; female students: β = −0.033, *t*-value = −0.512, *p* > 0.05). Thus, Hypothesis 6a was accepted. In addition, the negative impact of food insecurity on self-efficacy was greater among male students (β = −0.314, *t*-value = −3.506, *p* < 0.001) than female students (β = −0.059, *t*-value = −0.955, *p* > 0.05). Thus, Hypothesis 6b was also accepted. However, Hypothesis 6c was rejected because no significant difference was found in the effect of food insecurity on life satisfaction by gender. Based on these findings, the negative impact of food insecurity perceived by university students on their psychological well-being and self-efficacy was greater among male students. This result suggests that stronger awareness of food insecurity lowers an individual’s psychological well-being and confidence and that this negative impact is greater among male students.

## 6. Discussion and Implications

### 6.1. Conclusions and Discussion

The aim of this study is to explore the relationships between food insecurity, psychological well-being, self-efficacy, and life satisfaction among university students in South Korea. Our findings suggest that students’ perceived food insecurity negatively affects the status of psychological well-being. This result is consistent with preceding studies, which have confirmed that students’ food insecurity causes serious damage to their mental stability [110,139], and it exacerbates their psychological disorder problems by severely interfering with their everyday lives [82,138,140]. This finding also coincides with prior research results, which state that many students are suffering from serious psychological distress derived from food insecurity since the COVID-19 pandemic [108,109], and this tendency was particularly stronger in students who were under severe food restraints with financial vulnerability and inaccessibility to healthy foods [141,142]. Moreover, not only students’ psychological well-being but also self-efficacy is deeply associated with the notion of food insecurity [28,122,123,143]. It reconfirms previous research that states that an individuals’ food insecurity situation has a great detrimental effect on psychological aspects as well as cognitive skills such as self-efficacy, self-esteem, and self-confidence [35,112].

However, contrary to expectations, perceived food insecurity has no negative effects on students’ life satisfaction. Although this result is different from the majority of the prior studies that verified adverse effects of food insecurity on individuals’ life satisfaction [32,50,78,137,144], it is consistent with earlier studies conducted among university students who have other distinct important priorities such as housing, financial situation, or human relations rather than food security status [161]. Since the primary factors influencing individual life satisfaction may differ by each country [162] and generation [49,50], an analysis considering cultural and environmental contexts is necessary. Future qualitative studies are critically needed to clarify which living condition sources or personal factors, rather than food security, contribute to determining students’ overall contentment with life.

In addition, the level of students’ psychological well-being positively influences their life satisfaction [60,146]. One of the key results from this study is that, among research variables, psychological well-being has the largest influence on students’ life satisfaction. These findings are consistent with previous studies that confirm that psychological well-being and life satisfaction are inextricably linked to each other [163,164].

However, unlike the close relationship between psychological well-being and life satisfaction, not statistically significant results were obtained for the relationship between self-efficacy and life satisfaction. Although many researchers claim that one’s self-consciousness such as self-efficacy and self-esteem has a major impact on one’s quality of life by determining life satisfaction levels [122,135], no similar results were found in this study sample. This result indicates that self-efficacy is not an influential predictor of university students’ life satisfaction, and there is a need to closely examine other significant factors that directly affect students’ daily lives from a practical point of view. As mentioned earlier, life satisfaction factors may vary depending on the situation of each country or each age group, and individuals derive satisfaction from different sources based on their personal code of values [161]. In this sense, more studies need to be continued in the future to explore complex relationships between students’ life satisfaction and the potentially influential predictors of them.

Concerning the moderating effects of gender in this study, the results partially supported hypothesis 6, because the negative impact of food insecurity on psychological well-being and self-efficacy was greater in male students rather than female students. This finding is similar with preceding studies, which indicate that male students tend to be more conscious of their food insecurity status [56] and sustain a lot of mental traumas from their precarious living situations [165]. Although there are prior studies reporting that females are more vulnerable to food insecurity [150], our findings contradict this, and one possible explanation could be that males are more inexperienced in solving their food insecurity status and have more difficulty coping with poor diets than females who generally have more knowledge about skillfully managing their diet plans [56]. In addition, another plausible reason is that each research may achieve different results depending on the national and cultural circumstances or other extraneous contextual factors in which the respondents are placed [128]. Further exploration studies are needed to confirm these results and establish the primary causes of gender difference. Therefore, all the research variables in this study should be underscored as important factors, and further extended studies must be carried out to provide better insight concerning students’ food insecurity status. It is expected that future studies with more macroscopic and holistic views will be needed to derive more valuable information for the enhancement of students’ food security.

### 6.2. Implications for Research and Practice

Since food dominates one’s life, food insecurity is a very significant issue that needs to be urgently addressed. Through previous studies and the results presented in this research, it has clearly been shown that food insecurity affects university students’ psychological well-being and self-efficacy, and side effects of food insecurity impose serious negative barriers on students’ daily lives. Since students’ academic success and overall well-being cannot be achieved when they are under a lot of psychological stress derived from poor and irregular eating habits, it is necessary to find effective ways to solve food insecurity problems. Based on these findings, this study suggests multiple significant implications for university, government, and local organizational institutions.

First, universities should pay more attention to the issue of students’ food insecurity and further accelerate related research from various angles. To be more specific, not only qualitative approaches but also preemptive and qualitative research methods will be helpful for closely examining students’ present food security situation and hearing their personal thoughts on related issues. In particular, the in-depth interview method will be the surest way for discovering students’ true feelings about their food insecurity status by exploring the primary factors of food security such as high food prices, raises in tuition fees, low income, or inadequate financial aid. This will also be effective in the finding of key solutions to fundamental problems that arise in each student’s situation. In addition, dividing students into subgroups according to their food security levels based on standardized criteria categories may be an efficient way to find suitable solutions for each group [79]. Identifying distinct characteristics of different groups and discovering their common features will be a great help to researchers for looking for appropriate solutions for each group and suggesting implications for reducing food insecurity severities.

Secondly, in a more practical sense, universities should provide student assistance programs and internally create stable food environments for the enhancing of students’ food security. For example, by conducting systemic food literacy education or nutritional classes as part of the required curriculum, universities can help students recognize the importance of ideal eating habits by guiding them on how to maintain sustainable healthy diets even with limited resources. In order to successfully implement these educational intervention strategies, exploiting media tools and developing online educational programs will be very effective in increasing students’ accessibility so that many students can easily utilize educational programs more often, without restrictions on time and space. According to related prior studies, the fact that many students want this type of university food-related education and additional outreach for alleviating their food insecurity was established [142], and it was proven that students succeeded in making ideal food choices and maintaining healthy eating habits through these educational programs [54]. However, universities’ systematic education about food security were found to be insufficient in the case of South Korea when comparing with that of Western nations. Therefore, educators and supporters who are engaged in Korean universities should lead stronger efforts to prepare structured educational environments and provide systematic programs with supportive services. If additional longitudinal studies that verify the short-term and long-term effectiveness of food security educations on students are conducted, universities’ educational programs will become strengthened, progressively improving students’ food security status and promoting their well-being.

Furthermore, in addition to micro level implications for university institutions, as a measure of precaution against aggravating students’ food insecurity status, more solutions from a macroscopic perspective must be prepared as well. For instance, universities can provide various campaign activities such as farmer’s market or campus food pantries in concert with local organizations. This type of national and community level support will promote both student awareness of food security issues and the possibility of purchasing healthy ingredients at a reasonable price. In addition, public health professionals should simultaneously work with university institutions to screen students’ food security in a more systematic manner in order to suggest reasonable policies for improving student welfare. These collective efforts and integrative cooperation among various actors in society will greatly assist food insecure students in need of immediate financial support and practical help. All these holistic approaches and social cohesions for alleviating multi-faceted risk factors that threaten students’ food security are expected to contribute to proposing sustainable eating plans over the long run.

### 6.3. Strengths and Limitations

To the best of our knowledge, this study is the first research work that thoroughly focuses on the effects of university students’ food insecurity on their psychological states and life satisfaction in Republic of Korea during the COVID-19 pandemic. Moreover, unlike many previous studies that solely address the subject of food insecurity with physical health, this research deeply examines the potential effects of perceived food insecurity on various psychological and cognitive aspects from the perspective of university students. In addition, it incorporates a unique research concept, which is self-efficacy, into the food insecurity issue and also explored gender differences to provide more interesting implications. Thus, this research is considered meaningful in terms of expanding the scope of existing research by providing valuable research data with novel viewpoints. Therefore, further extended studies with more macroscopic and holistic views are needed to derive more valuable information and to provide better insight into the enhancement of students’ food security.

Even though this study includes important considerations and future implications from various perspectives, it must be acknowledged that it still has several limitations. First, the respondents to the survey are limited to university students in South Korea, which means that it is difficult to generalize the results for people of all ages around the world. Therefore, additional meaningful results will be obtained if follow-up research is conducted targeting more diverse cultural groups. Second, because this study only used quantitative research methods, there is a possibility that the psychological aspects of the respondents were not sufficiently reflected. Therefore, if qualitative investigations such as focus group discussions or one-on-one in-depth interviews are conducted, researchers may obtain more detailed information from respondents. Third, this study solely examines the moderating effects of gender differences. In order to achieve specific results from more expanded perspectives, it is imperative to verify the moderating effects of other influential demographic factors, such as students’ grades, food cost standards, and dwelling environments. Lastly, since this study only focuses on food insecurity with several psychological domains and the cognitive aspects of students, future studies can expand the scope of the research by combining both physical outcomes and psychological consequences with students’ food insecurity status.

## Figures and Tables

**Figure 1 foods-12-03429-f001:**
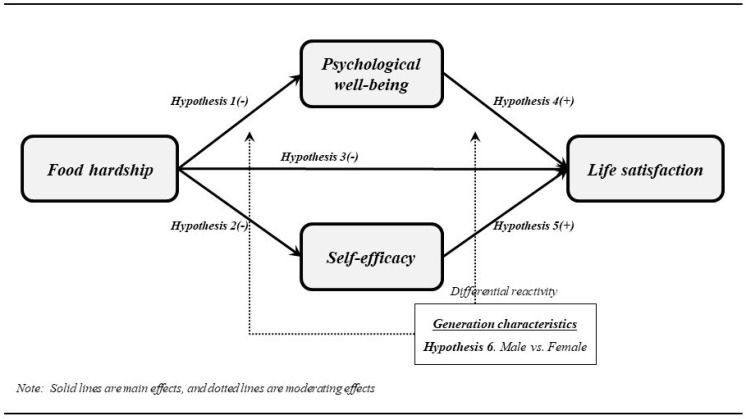
Research model.

**Table 1 foods-12-03429-t001:** Confirmatory factor analysis and reliability analysis.

Construct (Cronbach’s Alpha)	Standardized Estimate	*t*-Value	CCR ^a^	AVE ^b^
Food hardship (0.870)			0.903	0.531
FH_1_	0.713	fixed		
FH_2_	0.746	15.120 ***		
FH_3_	0.769	15.537 ***		
FH_4_	0.744	15.077 ***		
FH_5_	0.674	13.726 ***		
FH_6_	0.725	14.717 ***		
Psychological well-being (0.886)			0.876	0.610
PW_1_	0.783	fixed		
PW_2_	0.728	16.920 ***		
PW_3_	0.760	17.820 ***		
PW_4_	0.828	19.816 ***		
PW_5_	0.803	19.084 ***		
Self-efficacy (0.918)			0.871	0.738
SE_1_	0.881	fixed		
SE_2_	0.845	24.908 ***		
SE_3_	0.849	25.118 ***		
SE_4_	0.862	25.831 ***		
Life satisfaction (0.926)			0.866	0.808
LS_1_	0.940	fixed		
LS_2_	0.826	27.041 ***		
LS_3_	0.928	35.956 ***		

Note: ^a^ CCR = composite construct reliability; ^b^ AVE = average variance extracted; Standardized estimate = β-value; χ^2^ = 497.979 (df = 129) *p* < 0.001; χ^2^/df = 3.860; Goodness of Fit Index (GFI) = 0.894; Tucker–Lewis Index (TLI) = 0.929; Comparative Fit Index (CFI) = 0.940; Incremental Fit Index (IFI) = 0.940; Root Square Error of Approximation (RMSEA) = 0.076; RMR = 0.054; *** *p* < 0.001.

**Table 2 foods-12-03429-t002:** Means, standard deviations, and correlation analyses.

Construct	1	2	3	4	5	Mean ± SD ^a^
1. Gender	1					1.68 ± 0.468
2. Age	−0.139	1				22.52 ± 2.116
3. Food hardship	−0.024	−0.064	1			4.67 ± 0.661
4. Psychological well-being	−0.008	−0.110 *	−0.118 **	1		5.06 ± 0.993
5. Self-efficacy	−0.059	−0.023	−0.127 **	0.702 **	1	4.73 ± 1.154
6. Life satisfaction	−0.065	−0.045	−0.104 *	0.693 **	0.592 **	4.43 ± 1.298

Note: ^a^ SD = Standard Deviation; All variables were measured on a 7-point Likert scale from 1 (strongly disagree) to 7 (strongly agree), * *p* < 0.01; ** *p* < 0.05.

**Table 3 foods-12-03429-t003:** Structural parameter estimates.

Hypothesized Path (Stated as Alternative Hypothesis)	Standardized Path Coefficients	*t*-Value	Results
H1: Food hardship → Psychological well-being	−0.129	−2.476 *	Accepted
H2: Food hardship → Self-efficacy	−0.144	−2.814 **	Accepted
H3: Food hardship → Life satisfaction	−0.029	−0.085	Rejected
H4: Psychological well-being → Life satisfaction	0.715	9.566 ***	Accepted
H5: Self-efficacy → Life satisfaction	0.080	1.189	Rejected
Goodness-of-fit statistics	χ^2^_(127)_ = 372.917 (*p* < 0.001)		
	χ^2^/df = 2.936		
	GFI = 0.924		
	IFI = 0.960		
	CFI = 0.960		
	RMR = 0.048		
	RMSEA = 0.063		

Note: * *p* < 0.05, ** *p* <0.01, *** *p* < 0.001; GFI = Goodness of Fit Index; NFI = Normed Fit Index; CFI = Comparative Fit Index; RMSEA = Root Mean Square Error of Approximation.

**Table 4 foods-12-03429-t004:** Moderating effects on university students’ gender.

Hypothesized Path	Male (*n* = 159)		Female (*n* = 332)		Constrained Model χ^2^ (df = 255)	∆χ^2^ (df = 1)
	Standardized Coefficients	*t*-Value	Standardized Coefficients	*t*-Value		
H1: Food hardship → Psychological well-being	−0.307	−3.347 ***	−0.033	−0.512 ns	522.241	6.53 *
H2: Food hardship → Self-efficacy	−0.314	−3.506 ***	−0.059	−0.955 ns	520.382	4.68 *
H3: Food hardship → Life satisfaction	−0.063	−1.051 ns	−0.007	−0.163 ns	516.216	0.51 ns
H4: Psychological well-being → Life satisfaction	0.475	3.889 ***	0.812	8.806 ***	522.924	7.22 *
H5: Self-efficacy → Life satisfaction	0.345	2.909 **	−0.030	−0.376 ns	522.373	6.67 *

Note: Unconstrained model χ^2^ = 515.702; df = 254; GFI = 0.898, CFI = 0.958; AIC (Akaike Information Criterion) = 691.702; * *p* < 0.05, ** *p* < 0.01, *** *p* < 0.001, ns—not significant.

## Data Availability

The data presented in this study are available on request from the first author.

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
