# Peer review of "Food Insecurity during the Pandemic in South Korea: The Effects of University Students’ Perceived Food Insecurity on Psychological Well-Being, Self-Efficacy, and Life Satisfaction"

_foods, 2023, doi:10.3390/foods12183429_

Round 1

Reviewer 1 Report

In this manuscript, the authors are pointing a global challenging issue in the public health: Food insecurity.

The authors did a good job. The manuscript is clearly written, and all relevant tables are clear for the reader.

Author Response

Thank you so much for the nice answer.

Reviewer 2 Report

This is a well-written manuscript with a novel interesting topic. Several comments may promote it:

- Better to explain about food insecurity in hard situations such as disasters and refer to some published relationships (e.g. BMJ Mil Health. 2021 Jun;167(3):153-157. doi: 10.1136/jramc-2019-001301. Epub 2020 Feb 20.)

 - It better to check whole the manuscript for grammatical and typo errors. Maybe involving a native English person. 

- Discussion: Type of the study may cause bias. Please mention it as a limitation. 

 - It better to check whole the manuscript for grammatical and typo errors. Maybe involving a native English person. 

Author Response

Thank you for your careful review.
1. The introduction has been added with information on food insecurity.
2. The manuscript was proofread by a native speaker twice before submission.
3. These details have been added to the limitations.

Reviewer 3 Report

I think this study is very interesting and intriguing. It is nice to read about this kind of studies that integrate psychological perspectives with other disciplines. I appreciate the choice of the authors and the work they have done. 

However, I think the article does not present a very good structure and the level of the language is quite low. For publication, I think the authors should go with proofreading with expertise in psychology to improve the quality of the manuscript. Also, the authors should not report the type of tools used for analysis (e.g., AMOS) but rather the type of analyses they have done. 

 I think the authors should go with proofreading with expertise in psychology to improve the quality of the manuscript. The contents are good and interesting but their presentation is not very professional.

Author Response

Thank you for your review.
This study applies native-speaker proofreading to UK-based and US-based investigators with two changes before submission.
Additional mention of the research method (type of analyze) was made in the manuscripts.